# Two surveys separated by almost a decade reveal the inexorable decline in sleep time in teens: Results of the National survey of middle and high schools in adolescents on health and substances (EnCLASS 2018), and the evolution since 2010

Damien Léger[1,2]*, Virginie Ehlinger[3], Stanislas Spilka[4,5], Olivier Le-Nézet[4], Brigitte Fauroux[1,6], Victor Pitron[1,2], Emmanuelle Godeau[3,4,5,7]

1 Université Paris Cité, VIFASOM (UMR Vigilance Fatigue Sommeil et Santé Publique, Paris, France, 2 APHP Hôtel Dieu, Centre du sommeil et de la vigilance, Paris, France, 3 CERPOP-UMR1295, unité mixte INSERM, Université Toulouse III Paul Sbatier-Equipe SPHERE, Toulouse, France, 4 Observatoire français des drogues et toxicomanies (OFDT), Paris, France, 5 faculté de médecine, faculté de médecine UVSQ, Université Paris-Sud, Université Paris-Saclay, Inserm, CESP, Villejuif, France, 6 Unité d'exploration respiratoire du sommeil des enfant, APHP, Necker, Paris, France, 7 École des hautes études en santé publique (EHESP), Rennes, France

* damien.leger@aphp.fr

## Abstract

### Purpose

We investigated total sleep time (TST) and sleep paradigms during schooldays and leisure days in teenagers, and compared the findings to results from 8 years before.

### Methods

The EnCLASS epidemiological survey is a representative sample of thousands of French middle and high schoolers. Specific sleep questions allowed us to assess TST, sleep debt, too short sleep, and screen use. Sleep debt as was defined as a difference between the total sleep time (TST) during schooldays (TSTS) non-schooldays (weekends or vacations; TSTN) of over 2 hours. Too short sleep was assesses when TSTS was < 7 hours. We detailed results of the 2018 bivariable analysis and adjusted by sex, age, class level, academic delay and socio-economic level, allowing us to compare the 2018 and 2010 results.

### Results

TST decreases constantly from the beginning of middle school (MS) to the end of high school (HS). In 2018, 26.7% of middle and 43.7% of high schoolers were in "sleep debt". Additionally, 13.8% of middle and 29% of high schoolers reported too short sleep (less than 7 hours) on schooldays (vs. 7.8% in middle and 25.1% in high school in 2010). In 8 years, the TSTS of middle schoolers decreases by an average of 20 minutes of sleep per

**Data availability statement:** The data would be available only upon request. Data cannot be shared because the study has interviewed minors under 18 of age in colleges and high school, their data, are protected by the law and cannot be shared publicly. We did not get the approval of parents for sharing their children personal data publicly. Moreover authors do not have the permission to publicly share the data which are owned by the OFDT. However, with regard to the readers of PLOS-One, we undertake to respond on request addressed to the first author at damien.leger@aphp.fr to researchers wishing to share data from this study or to compare their own results for a scientific purpose compatible with our own and with a specific and limited agreement of confidentiality. The authors will then submit to the OFDT Scientific Council a request for the transmission of anonymized data for this scientific purpose, with the support of our author Stanislas Spilka stanislas.spilka@ofdt.fr. We also recommend that researchers contact the Observatoire français des drogues et toxicomanies (OFDT), Service enquêtes statistiques. https://www.ofdt.fr/. The authors undertake to facilitate any request for transmission of anonymized data for a scientific purpose compatible with ours, while respecting the rules of confidentiality and privacy promised to the parents of young children as a token of their full participation in the study.

**Funding:** The French inistry of Health, Santé Publique France (SPF) (https://www.santepubliquefrance.fr/) supported this paper and the research reported therein.

**Competing interests:** The authors declare no conflict of interest regarding these surveys and the manuscript.

night on weekdays, dropping from 8 h 35 min to 8 h 14 min. Regarding screens at bedtime, an increase of 36.8 points was observed on schooldays in MS between 6th and 9th grade, with 13.9% more in HS, in 12th grade. This included 40.6% of middle and 60.6% of high schoolers using internet before sleep on schooldays.

## Conclusion

Faced with this trend, teachers and parents need to take preventive action to avoid an inexorable decline in teenagers' sleep.

## Introduction

A good night's sleep is well recognized as a crucial factor in psychological cognition and emotion [1], and as a physical factor in physiological balance [2–3]. Maintaining a regular sleep schedule is also essential for school performance and societal accomplishment among young adults [4]. However, sleep can be compromised at night by activities such as school tasks, social networking on the internet, entertainment (music, movies and series), and the continuous checking of information and messages. These activities are a major reason for the progressive development of "sleep debt" among children and teenagers in recent years [5–6].

Based on data collected from October 1, 2010 to June 30, 2011, we document here for the first time the sleep of French adolescents using the Health Behaviour in School-aged Children (HBSC/WHO) survey [7–8] and the 2011 (April 1 to June 30) European School Survey Project on Alcohol and Other Drugs (ESPAD) survey [6]. The data reveal a growing proportion of middle schoolers with sleep debt and too short sleep during schooldays. Facing this evolving adolescent way of life and new uses of screens, we aimed to test students again and to compare their sleep using the same questionnaires. This was possible with the new EnCLASS 2018 (April 1 to June 30) survey, which allowed us to shed light on the sleep habits of middle school (MS) and high school (HS) students, their screen use in the evening, and their evolution in a representative sample of French middle schoolers.

## Matérials and methods

### Design

This work aims to compare the datasets of two previous surveys devoted to sleep habits and paradigms in middle and high schoolers, with a survey performed 8 years later and using the same methodology.

We conducted the first survey in 2010 for middle schoolers, with a second survey in 2011 for high schoolers (http://espad.org/ and http://hbsc.org/) [5–6]. The 2018 data are from the National Survey of Adolescent Health and Substances (EnCLASS), a merging of HBSC and ESPAD in France [7].

In order to attain nationally representative samples of students, the three surveys used a multi-stage balanced sampling, in which schools were randomly sampled at the first stage, and 2 classes were then selected from each at the second stage [8]. In 2010 and 2011, the balancing variables were the urban level of the city where the school was located, the academic class level, and the private/public status of the school. In 2018, the balancing variables were the urban level of the city where the school was located and whether it was located in a deprived area (yes/no). All students in the selected classes were invited to participate. The statistical department of the French Ministry of National Education (DEPP) drew the samples, in accordance with international requirements [9]. The administration of the survey followed identical

procedures: students anonymously self-completed the questionnaires at school during their class, within one hour, and with the supervision of an adult to guarantee confidentiality.

In 2010–2011, response rates in middle schools were 93.4% at the class level and 83.8% at the student level, while the rates were respectively 86.5% at the high school level. For the 2018 data, the final samples consisted of 367 MS and 275 HS at the national level, from which a representative sample of 7,659 middle and 6,048 high schoolers was drawn (with 90.5% at the class level and 87.2% at the student level in MS, and respectively 84.5% and 78.4% in HS). Non-responses were mainly due to the absence of students on the day of the survey, or more rarely due to refusal of participation by children or their parents (14% in total), or even schools (n = 56). The final sample consisted of 12,973 middle schoolers (from 308 MS) and 7,155 high schoolers (from 206 HS).

## Ethics

In 2010–2011, the study protocol was approved by the Ministry of Education and the French National Commission on Information Technology and Liberties (Commission Nationale Informatique et Libertés; CNIL). Similarly, the EnCLASS 2018 survey received a notice of opportunity from the National Council for Statistical Information (Cnis, n°142/H030), and was reported to the CNIL (under the reference 2155714 v 0), using the same process regarding parental authorization. An information letter was given to parents to explain the study's broad objectives, methods, content, and references. Children gave their oral consent. Parents who refused the participation of their child had to complete a written document and return it to their child's school, whereas no reply was necessary if they consented (passive consent). The ethics committees of the Ministry of Education approved this consent procedure.

## Sleep assessments

We introduced questions investigating sleep habits in the French HBSC survey for the first time in 2010 [5, 10]. These sleep-specific measurements were inspired by validated tools recommended for assessing sleep in adolescents, such as enquiring specifically and separately about sleep on school nights and weekends, and the use of diaries to keep a log of sleep hours.

## Sleep values

We calculated an estimate of the total sleep time (TST) during schooldays (TSTS) and non-schooldays (weekends or vacations; TSTN) based on the following questions:

1) 'When you have/don't have class the following day, at what time do you usually fall asleep?' with 13 possible answers ranging from "before 9 p.m." to "3 a.m. or later".

2) 'When you have/don't have class the following day, at what time do you usually wake up?' with 15 possible answers ranging from "before 5 a.m." to "after noon".

3) 'Usually how long does it take for you to fall asleep?' with 5 possible answers: "less than 10 minutes", "from 11 to 20 minutes", "from 21 to 30 minutes", "from 31 to 40 minutes", and "more than 40 minutes".

TST was defined as the difference between the time at which the participant went to bed and the time of day that they woke up, discounting the time needed to fall asleep.

## We defined sleep paradigms as follows:

- *Sleep debt*: Despite the absence of a consensual definition for sleep debt in adolescents, most authors and the National Sleep Foundation (USA) consider that a 2-hour debt is sufficient

to evoke sleep debt in teenagers [11–14]. We therefore defined sleep debt as a difference between TSTN and TSTS of over 2 hours.

- *Too short sleep*: In adults, subjects sleeping less than 6 hours during weekdays are usually considered "short sleepers" and may potentially be at a higher risk of developing comorbidities [15–17]. In teenagers, based on previous recommendations and observations, we considered that sleeping less than 7 hours was "too short" [16, 18, 19].

### Adjustment factors

We collected the following factors in the questionnaire: 1) sex; 2) year of birth, from which we could determine if the student is academically delayed (we considered children delayed when their age at testing was higher than the theoretical age for their class level); 3) grade, from 6th to 12th; 4) the type of education of high schoolers (general or vocational training); and 5) the socio-economic level, evaluated by the HBSC Family Affluence Scale (FAS), allowing to classify the respondents into 3 groups according to the scores: "Low", which includes the 20% of middle schoolers with the lowest scores; "Middle"; and "High", which contains the 20–30% of students with the highest scores.

Screen use: In 2018, middle and high schoolers were asked the following about screen activities in their bedroom just before falling asleep on schooldays and leisure days: "What do you most often do in bed just before falling asleep?" Possible answers included:

- You watch television, a film, a series;

- You play on a screen device (computer/console/tablet/smartphone);

- You go on the internet (computer/console/tablet/smartphone);

- You communicate on your phone/smartphone/tablet. You chat with loved ones/family.

### Statistical analyses

To be representative of the French teenage student population, data collected were weighted using national data concerning sex, age and student academic level distributions [20]. The samples of middle and high schoolers were described by numbers and frequencies for the categorical variables (gender, age, level, academic delay, socio-economic status), and by mean and standard deviation (SD) for the quantitative variable (age).
We then considered the following outcomes.

- TST during schooldays (TSTS) and non-schooldays (weekends or vacations; TSTN), using quantitative variables and linear regression.

- Sleep debt and too short sleep, using binary variables and logistic regression.

For each outcome, we performed a bivariate description in order to estimate the unadjusted difference between the two periods, followed by a regression model adjusted for sex, class level, academic delay, family affluence, and type of education (in high school). The non-independence of the observations was taken into account in the bivariate (SURVEY commands under SAS, SVY commands under Stata; tests with Rao Scott's correction for categorical) and multivariate analyses by applying generalized estimating equation (GEE) models. Statistical tests were applied at a threshold of 5%.

### Results

From October 1, 2010 to April 30, 2011, 13,707 secondary school students completed the questionnaires, including 7,659 middle and 6,048 high schoolers (50% boys). In 2018 (from

April 1 to June 30), the final sample comprised 20,128 students (12,973 middle and 7,155 high schoolers, with 50.6% boys).

## Total sleep time (TST)

TST during non-schooldays (weekends or vacations; TSTN) and schooldays (TSTS) within the two periods (2010–2011, 2018) are presented and compared in "Table 1".

- In both periods, TST during school days (TSTS) significantly decreased grade by grade in MS, with an average reduction in TSTS by almost one hour between students in the 1st year of MS and students in the last year of MS. Moreover, middle schoolers in the highest grades had lower TST during leisure days (TSTN) than middle schoolers from the lowest grades, with average differences of 37 minutes in 2010 and 36 minutes in 2018.

**Table 1. Total sleep time (TST) on non-schooldays (weekends or vacations; TSTN) and on schooldays (TSTS), for middle and high schools in 2010-11 vs. 2018 (bivariate analyses).**

| | TSTSN | | | | | | | | TSTS | | | | | | | |
|---|---|---|---|---|---|---|---|---|---|---|---|---|---|---|---|---|
| | MS 2010 | | HS 2011 | | MS 2018 | | HS 2018 | | MS 2010 | | HS 2011 | | MS 2018 | | HS 2018 | |
| | Mean | SD | Mean | SD | Mean | SD | Mean | SD | Mean | SD | Mean | SD | Mean | SD | Mean | SD |
| | 9:59 | 1' | 9:15 | | 9:35 | 2' | 9:18 | | 8:35 | 2' | 7:24 | | **8:14** | 2' | 7:19 | |
| **Grade** | | | | | | | | | | | | | | | | |
| 6th | 10:14 | 2' | - | - | 9:53 | 2' | - | - | 9:10 | 2' | - | - | 8:49 | 2' | - | - |
| 7th | 9:59 | 3' | - | - | 9:43 | 3' | - | - | 8:48 | 2' | - | - | 8:27 | 2' | - | - |
| 8th | 9:56 | 2' | - | - | 9:27 | 2' | - | - | 8:24 | 3' | - | - | 7:56 | 3' | - | - |
| 9th | 9:47 | 2' | - | - | 9:17 | 3' | - | - | 8:00 | 3' | - | - | 7:42 | 3' | - | - |
| 10th | - | - | 9:16 | 2' | - | - | 9:20 | 2' | - | - | 7:26 | 2' | - | - | 7:26 | 2' |
| 11th | - | - | 9:19 | 4' | - | - | 9:20 | 2' | - | - | 7:22 | 3' | - | - | 7:18 | 2' |
| 12th | - | - | 9:07 | 3' | - | - | 9:14 | 2' | - | - | 7:23 | 3' | - | - | 7:13 | 2' |
| Terminal | - | - | 9:07 | 3' | - | - | 9:14 | 2' | - | - | 7:23 | 3' | - | - | 7:13 | 2' |
| *P-value* | *<0.001* | | *0.150* | | *<0.001* | | *0.168* | | *<0.001* | | *0.231* | | *<0.001* | | *0.006* | |
| **Gender** | | | | | | | | | | | | | | | | |
| Boys | 9:45 | 2' | 9:02 | 2' | 9:24 | 2' | 9:05 | 2' | 8:38 | 2' | 7:19 | 2' | 8:17 | 2' | 7:14 | 2' |
| Girls | **10:12** | 2' | 9:27 | 2' | **9:47** | 2' | **9:31** | 2' | 8:33 | 2' | 7:28 | 2' | 8:10 | 2' | **7:25** | 2' |
| *P-value* | *<0.001* | | | | *<0.001* | | *<0.001* | | *0.013* | | | | *<0.001* | | *<0.001* | |
| **Academic delay** | | | | | | | | | | | | | | | | |
| No‡ | 10:02 | 1' | 9:21 | 3' | 9:37 | 2' | 9:21 | 2' | 8:40 | 2' | 7:30 | 2' | 8:16 | 2' | 7:22 | 2' |
| Delayed | 9:42 | 3' | 9:05 | 2' | 9:18 | 3' | 9:10 | 3' | 8:14 | 3' | 7:14 | 1' | 7:53 | 3' | 7:12 | 3' |
| *P-value* | *<0.001* | | | | *<0.001* | | | | *<0.001* | | | | *<0.001* | | | |
| **Socio-economic level FAS †** | | | | | | | | | | | | | | | | |
| Low | 9:48 | 3' | 9:14 | 3' | 9:29 | 3' | 9:13 | 2' | 8:30 | 3' | 7:19 | 2' | 8:11 | 3' | 7:14 | 2' |
| Average | 10:03 | 2' | 9:13 | 3' | 9:36 | 2' | 9:17 | 3' | 8:37 | 2' | 7:22 | 2' | 8:14 | 2' | 7:19 | 2' |
| High | 9:56 | 2' | 9:17 | 3' | 9:36 | 2' | 9:22 | 2' | 8:36 | 2' | 7:29 | 2' | 8:15 | 3' | 7:23 | 2' |
| *P-value* | *<0.001* | | *0.12* | | *0.101* | | *0.01* | | *0.040* | | *0.01* | | *0.484* | | *<0.001* | |

Based on the HBSC 2010, ESPAD 2011 and EnCLASS 2018 data.

Abbreviations: SD = standard deviation; ' = minute; No

‡academic delay = Students of the right age or at least one year ahead; FAS = Family Affluence Scale, a metric of 4-6 items to estimate the socio-economic level of parents; MS = middle school; HS = high school.

- Compared to 2010, middle schoolers in 2018 had a significantly shorter TSTS with an average of 21 minutes difference every night, and a significantly shorter TSTN with an average of 24 minutes difference every leisure night.

- Compared to 2010, high schoolers did not significantly vary their TSTS or TSTN in 2018.

- Regarding TSTN, girls reported significantly more sleep on leisure days than boys during the two periods. Regarding TSTS, boys in MS and girls in HS slept longer.

- Academically delayed middle schoolers always had a shorter TST than other students.

- The academic level of parents was typically not associated with shorter TST in HS children.

## Sleep debt and too short sleep

The evolution of sleep debt and short sleep among middle and high schoolers during the two periods is displayed in "Fig 1" and "Fig 2", and "Table 2" (supplementary statistical information).

- High rates of sleep debt were seen in both MS (27.0%) and HS (43.7%) students in 2018, with comparable global percentages in 2010.

- In 2018, significantly less middle schoolers had sleep debt than high schoolers, as compared to 2011.

- Girls had significantly more sleep debt than boys, whatever their age, grade or the year of the survey. Girls in HS were the most concerned by this, with 46.5% showing sleep debt.

- We found a significant association between sleep debt and academic delay in MS students, both in 2010 and 2018.

- HS with parents having the highest socio-economic level (estimated by the FAS scale) had a significantly lower rate of sleep debt in HS students in 2011.

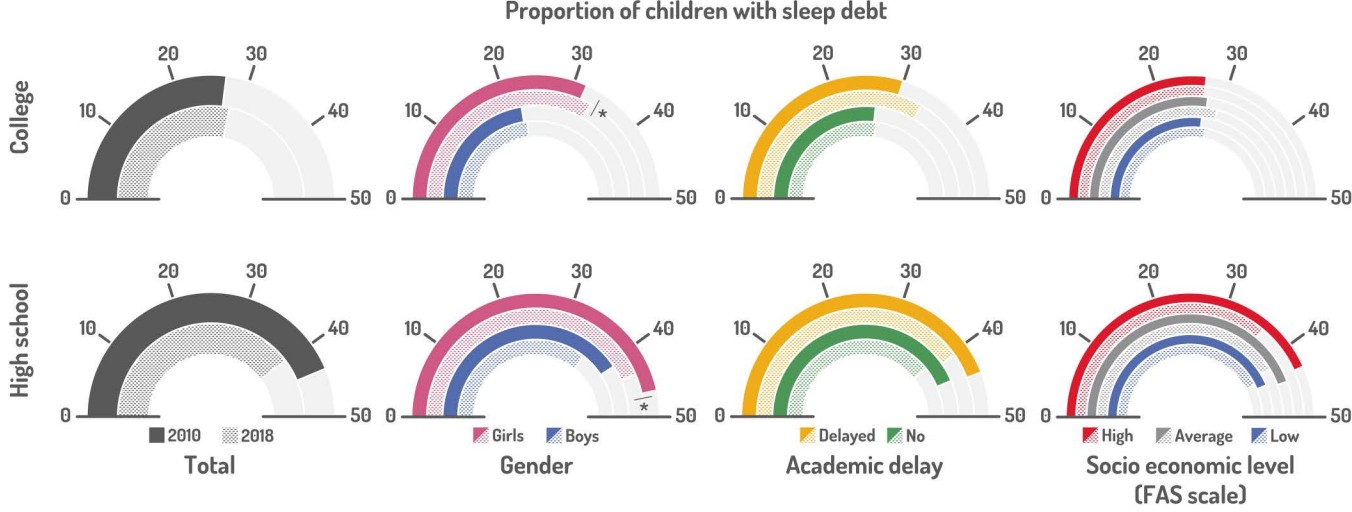

**Fig 1. Sleep debt as percentages of middle school and high schoolers in 2010-11 vs. 2018 (bivariate analyses).**

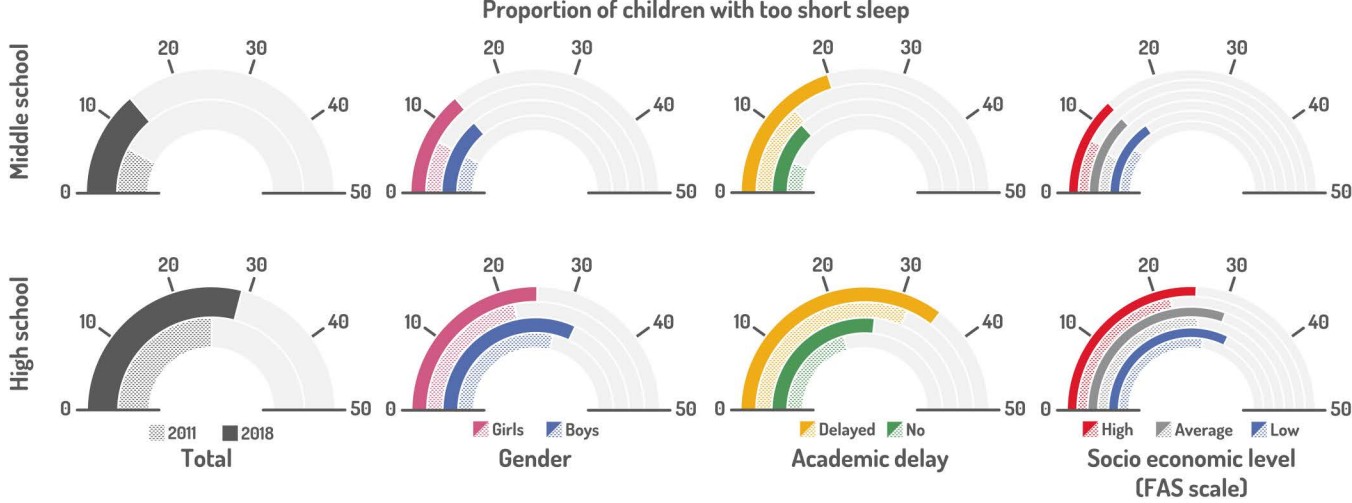

**Fig 2. Too short sleep as percentages of middle school and high schoolers in 2010-11 vs. 2018 (bivariate analyses).**

- The rate of children with too short sleep significantly increased in MS and HS between 2010-11 and 2018, from 7.8 to 13.8% in MS and from 25.1 to 29.0% in HS. The highest rate, 30.6%, was observed in 12th grade in 2018.

- Significantly more boys in HS are short sleepers than girls.

- Short sleep was significantly associated with academic delay in both MS and HS, in both periods.

**Table 2. Sleep debt and too short sleep as percentages of middle and high schoolers in 2010-11 vs. 2018 (bivariate analyses).**

| | Sleep debt | | | | Too short sleep | | | |
|---|---|---|---|---|---|---|---|---|
| | MS 2010 | HS 2011 | MS 2018 | HS 2018 | MS 2010 | HS 2011 | MS 2018 | HS 2018 |
| | % | % | % | % | % | % | % | % |
| **Total** | 28.2 | 39.7 | 27.0 | 43.7 | 7.8 | 25.1 | 13.8 | 29.0 |
| *P-value* | *<0.001* | *0.048* | *<0.001* | *0.213* | *<0.001* | *0.248* | *<0.001* | *0.097* |
| **Gender** | | | | | | | | |
| Boys | 23.2 | 35.6 | 22.5 | 40.8 | 7.8 | 28.6 | 13.7 | 32.0 |
| Girls | 33.3 | 43.5 | 31.7 | 46.5 | 7.8 | 21.8 | 13.9 | 25.2 |
| **Academic delay** | | | | | | | | |
| No‡ | 27.0 | 39.2 | 26.6 | 43.6 | 6.3 | 20.6 | 13.0 | 26.7 |
| Delayed | 33.5 | 40.3 | 29.8 | 44.1 | 13.7 | 31.4 | 20.1 | 35.4 |
| **Socio-economic level FAS †** | | | | | | | | |
| Low | 27.8 | 42.6 | 26.8 | 43.7 | 10.5 | 27.6 | 15.5 | 32.4 |
| Average | 29.3 | 41.3 | 27.4 | 44.4 | 6.9 | 26.2 | 13. | 30.4 |
| High | 26.8 | 36.1 | 26.8 | 43.3 | 7.7 | 22.1 | 13.1 | 25.7 |

Supplementary information for Figs 1 and 2.

Legend: Based on the HBSC 2010, ESPAD 2011 and EnCLASS 2018 sleep paradigms.

**Sleep debt**: We defined sleep debt in our subjects as a difference between TSTN and TSTS of over 2 hours.

**Too short sleep**: In teenagers, based on previous recommendations and observations, we considered that sleeping 7 hours or less was "too short".

Abbreviations: MS = middle school; HS = high school; % = percentage.

- The rate of short sleepers was significantly associated with the socio-economic level of parents estimated by the FAS scale.

## Screen use prior to sleeping and its impact on TST (2018 EnCLASS only)

Table 3 and Fig 3 show TST values by pre-sleep screen activity. Middle schoolers reporting pre-sleep screen activity had a lower TSTS, by 45-55 minutes on average, compared to peers who did not use a screen (30 minutes for those who watched films or series compared to those who reported not watching them). On days without class, the difference was around 25-37 minutes, and the TST of students who watched movies or series before sleeping did not differ from the sleep time of their peers at the 5% threshold.

The proportion of students reporting screen activities (computer/console/tablet/smartphone) before sleeping on evenings with or without class the next day increased along class years (p-values < 0.001). An increase of 36.8 points was observed in MS between the 6th and 9th grades when students had class the next day, while there was a 13.9% increase in HS students in 12th grade. This included 41.1% of middle and 48.8% of high schoolers watching television, a film or a series; 37.5% of middle and 39.8% of high schoolers playing on screens; 40.6% of middle and 60.6% of high schoolers going on the internet; and finally, 38.4% of middle and 66.0% of high schoolers communicating and chatting with loved ones/family. The rates were higher on evenings with no class the next day.

## Discussion

Our study was specifically designed to assess the epidemiology of TST in a large number of children and adolescents. Several authors have made a link between reduced sleep time and behaviors such as screen time or psychiatric and metabolic disorders (including obesity, depression-suicidality, fatigue, pain, and car accidents) [4, 21]. However, many researchers still use the single item "On average, how many hours of sleep do you get in a 24 h period?" to assess TST [22], or approximate items such as "How many hours do you sleep in bed on nights before schooldays" that are divided into < 8 hours and ≥ 8 hours per night [23]. Others have observed, from an epidemiological point of view, the impact of sleep timing (bedtime, wake-up time, and midpoint of sleep) on various health indicators. Interestingly, Dutil et al. conducted a systematic review on this topic, based on forty-six observational studies from 21 countries that included 208,992 unique participants. Sleep timing was assessed objectively using actigraphy in 24 studies, and subjectively in 22 studies [24]. Their findings suggest that later sleep timing is associated with poorer emotional regulation, lower cognitive function/ academic achievement, shorter sleep duration/poorer sleep quality, poorer eating behaviors, lower physical activity levels, lower quality of life/well-being, lower cognitive performance, and behaviors that are more sedentary. However, the authors regret that "none of these measurements of sleep timing account for sleep duration; in fact, very few studies adequately controlled for sleep duration in their analyses", and they postulated that "controlling for sleep duration is important given the well-established associations between short sleep duration with chronic diseases" [25].

One strength of our work is the comparison of average TST between grades from the beginning until the end of secondary studies, 8 years apart, with the same methodology and overall survey procedures, in wide, nationally representative samples. Our main finding is that middle school students have lost on average 20 minutes of sleep per night between 2010 and 2018.

Our work highlights another crucial issue, which is the very high and increasing level of "too short sleep" between 2010-11 and 2018, from 7.8 to 13.8% in MS and from 25.1 to 29.0%

**Table 3. Total sleep time (TST) on non-schooldays (weekends or vacations; TSTSN) and on schooldays (TSTS) for middle and high schools in 2018, according to screen activities before sleep.**

| Grade | TSTSN | | | | | | | | | TSTS | | | | | | | | |
|---|---|---|---|---|---|---|---|---|---|---|---|---|---|---|---|---|---|---|
| | 6th | 5th | 4th | 3rd | MS | 2nd | 1st | Term. | Total HS | 6th | 5th | 4th | 3rd | Total MS | 2nd | 1st | Term. | Total HS |
| | Mean (SD) | Mean (SD) | Mean (SD) | Mean (SD) | Mean (SD) | Mean (SD) | Mean (SD) | Mean (SD) | Mean (SD) | Mean (SD) | Mean (SD) | Mean (SD) | Mean (SD) | Mean (SD) | Mean (SD) | Mean (SD) | Mean (SD) | Mean (SD) |
| **Television, film, series** | | | | | | | | | | | | | | | | | | |
| No | 9:47 (3') | 9:42 (4') | 9:20 (4') | 9:13 (4') | 9:32 (2') | 9:14 (2') | 9:16 (2') | 9:08 (3') | 9:13 (2') | 8:53 (2') | 8:36 (2') | 8:07 (3') | 7:53 (3') | 8:26 (2') | 7:30 (2') | 7:26 (2') | 7:16 (2') | 7:25 (2') |
| Yes | 9:56 (2') | 9:43 (3') | 9:30 (2') | 9:18 (3') | 9:36 (2') | 9:22 (2') | 9:21 (2') | 9:16 (1') | 9:20 (1') | 8:39 (3') | 8:12 (4') | 7:42 (3') | 7:32 (3') | 7:56 (2') | 7:14 (2') | 7:05 (2') | 7:07 (2') | 7:09 (2') |
| P-value | | | | | 0.067 | | | | 0.009 | | | | | <0.001 | | | | <0.001 |
| **Playing on screens** | | | | | | | | | | | | | | | | | | |
| No | 10:10 (2') | 10:03 (3') | 9:48 (3') | 9:32 (3') | 9:55 (2') | 9:30 (2') | 9:25 (2') | 9:17 (2') | 9:24 (2') | 8:59 (2') | 8:42 (2') | 8:15 (2') | 7:55 (3') | 8:31 (2') | 7:33 (2') | 7:23 (2') | 7:19 (2') | 7:25 (2') |
| Yes | 9:33 (3') | 9:24 (3') | 9:14 (3') | 9:07 (3') | 9:18 (2') | 9:10 (3') | 9:14 (2') | 9:11 (1') | 9:12 (1') | 8:20 (5') | 7:59 (3') | 7:29 (4') | 7:27 (4') | 7:45 (2') | 7:07 (2') | 7:06 (1') | 6:59 (3') | 7:04 (2') |
| P-value | | | | | <0.001 | | | | <0.001 | | | | | <0.001 | | | | <0.001 |
| **Internet** | | | | | | | | | | | | | | | | | | |
| No | 10:06 (2') | 10:00 (3') | 9:49 (3') | 9:31 (5') | 9:55 (2') | 9:31 (3') | 9:27 (4') | 9:17 (3') | 9:26 (3') | 8:59 (2') | 8:43 (2') | 8:19 (2') | 8:02 (3') | 8:36 (2') | 7:41 (2') | 7:31 (1') | 7:28 (3') | 7:34 (2') |
| Yes | 9:29 (3') | 9:24 (3') | 9:14 (3') | 9:10 (3') | 9:18 (2') | 9:14 (3') | 9:16 (1') | 9:13 (1') | 9:14 (1') | 8:16 (5') | 7:56 (4') | 7:30 (4') | 7:28 (3') | 7:41 (2') | 7:09 | 7:06 | 7:03 | 7:06 (2') |
| P-value | | | | | <0.001 | | | | <0.001 | | | | | <0.001 | | | | <0.001 |
| **Communication/chatting** | | | | | | | | | | | | | | | | | | |
| No | 10:00 (2') | 9:52 (2') | 9:38 (3') | 9:28 (4') | 9:48 (2') | 9:26 (3') | 9:22 (3') | 9:11 (3') | 9:20 (3') | 8:57 (2') | 8:42 (2') | 8:19 (2') | 8:01 (3') | 8:35 (2') | 7:39 (2') | 7:34 (3') | 7:17 (3') | 7:32 (3') |
| Yes | 9:35 (4') | 9:30 (4') | 9:19 (3') | 9:11 (3') | 9:21 (2') | 9:16 (2') | 9:18 (1') | 9:15 (1') | 9:16 (1') | 8:18 (5') | 7:56 (4') | 7:27 (4') | 7:28 (3') | 7:40 (2') | 7:13 (2') | 7:07 (2') | 7:09 (2') | 7:09 (2') |
| P-value | | | | | <0.001 | | | | 0.175 | | | | | <0.001 | | | | <0.001 |

Based on the HBSC 2010, ESPAD 2011 and EnCLASS 2018 question: What do you most often do in bed just before falling asleep?

You watch television, a film, a series

You play on a screen (computer/console/tablet/smartphone)

You go on the internet (computer/console/tablet/smartphone)

You communicate on your phone/smartphone/tablet. You chat with loved ones/family

Abbreviations: SD = standard deviation; ' = minute; Term = terminal year; MS = middle school; HS = high school.

in HS. It is consensually admitted that short sleep in adolescents is associated with overweight and obesity in teens [19]. Specifically, Owens et al. reviewed the consequences of too short sleep in teens and reported the association with obesity [25]. The authors estimated that for each hour of sleep lost, the odds of being obese increased in adolescents by 80%. Short sleep is also associated with mood disorders, depression, suicidal ideation, and drowsy driving in adolescents [26]. In the span of several reviews and experimental studies, Short et al. explored the field of too short sleep on mood, cognitive function, and adaptive functioning, and explored the slow-down of spindles in sleep-restricted adolescents [27–30]. As in most of these works, we also observed a higher rate of short sleeping in boys.

One new yet highly concerning result of our survey is the higher prevalence of short sleep among academically delayed students. In Norway, Hysing et al. found in their survey of 7,798 students (16-19 years old), after adjusting for sociodemographic information, that short sleep duration and sleep deficit were the two sleep measures most linked to poor academic results [31]. Conversely, Arsanow et al. observed 700 adolescents in grades seven through the terminal year, and found that short TST was not associated longitudinally with changes in emotional and academic functioning [26].

The socio-economic level of parents was found to significantly affect the rate of short sleepers, which was higher among the most disadvantaged families, even more in 2018 as

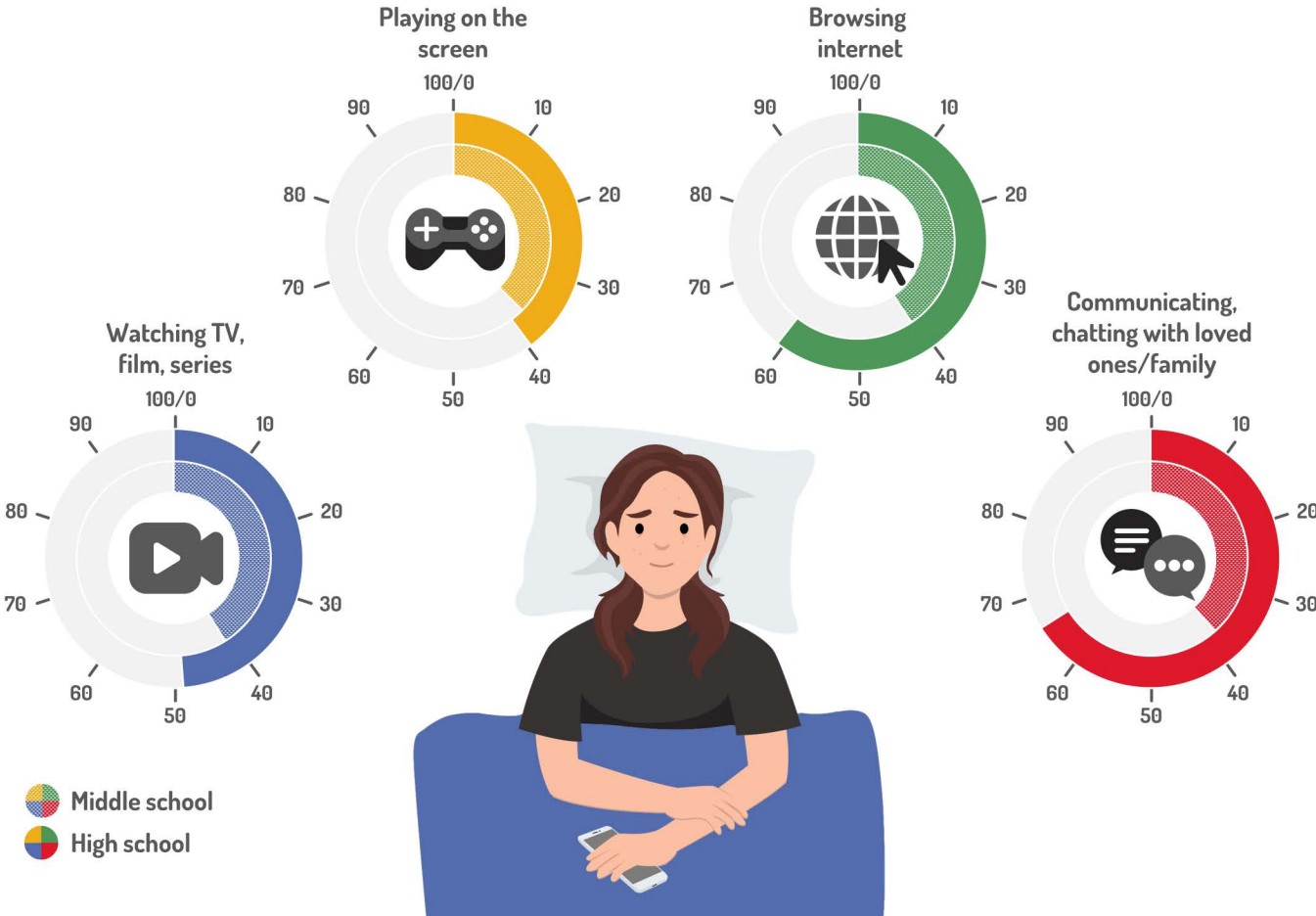

**Fig 3. Screen use prior to sleeping in percentages on schooldays and leisure days in middle and high schoolers in 2018.**

compared to 2010-11. Because we may consider too short sleep as a socio-economic indicator with a higher risk in vulnerable persons, this is an important public health concern that must be targeted and protected by specific prevention campaigns.

By definition, sleep debt results from short sleep on schooldays. An increase in the rate of short sleep increases the rate of sleep debt, which can be overcome if students are able to sleep more on leisure days. We have found that 27% of middle and 43.7% of high school French students suffer from sleep debt, with well-recognized consequences on cognition, mood and academic performance, as well as an increased risk of accidents [7]. In the adult population, we have shown that people under sleep debt compensated this deprivation by napping or increasing their sleep during the weekend [32]. Napping is a public health issue that directly concerns middle and high schools, where it is not possible to nap, even when students are severely sleep deprived [33]. In particular, we must better explain that napping has positive physiological, psychological and cognitive effects in schools [33–34]. An important factor relevant to prevention in our survey is that girls had significantly more sleep debt than boys did. Indeed, HS girls are the most concerned, with 46.5% of the group showing sleep debt. Fortunately, one survey of 205 16-17-year-old Australian teens (54.1% females) wearing actigraphs showed that sustained, cumulative perceived sleep debt during schooldays could be reduced by sleep during weekends and vacations. In particular, the trajectories of sleep debt during vacation suggest a recovery from school-related sleep restriction [35].

Finally, our survey provides valuable data for public health debates focused on screen use, sleep delay and sleep debt, and their impact on health and cognition [7, 36]. We only enquired about screens in 2018 with the question "What do you most often do in bed just before falling asleep?", which is very different from surveys with questions about screen use in the evening or during the day. Here, our focus is on the direct competition between sleeping and watching screens. This is why the high rate of children using screens in bed, and its dramatic increase with each grade, is of concern. We observed an increase of 36.8 points in MS, between 6th and 9th grade, when there is class the next day, with a 13.9% increase in 12th grade. Interestingly, we calculated how both general and specific uses of screens were associated with TST. The TSTS of middle schoolers reporting pre-sleep screen activity was 45-55 minutes shorter on average compared to peers who did not use any screens. The most deleterious screen activities in bed included surfing on the internet or communicating/chatting with friends and family. This is physiologically understandable, as the undesirable effects of screens on sleep may come from both the blue light that delays melatonin onset and the intellectual excitement related to receiving new information from others. To cope with the various effects of screen use, public health campaigns, educators and parents may try to differentiate the limiting of screen use in children more specifically [37,38].

We acknowledge several limitations in our work. First, this questionnaire survey only allows subjective data collection on a single day. Second, we did not ask any questions about napping time during weekends, which could have helped to better understand how children compensate for sleep debt outside of night sleep. Third, we did not interview children on how they perceived their sleep was too short or not. Finally, we do not have any details regarding parental involvement in sleep hygiene, or how parents themselves or the family model could affect the sleep habits of their children.

## Conclusion

Our work highlights that sleep duration among French secondary school students has declined, specifically in middle school, with high rates of too short sleep and sleep debt in both middle and high schools. Given these results, we hope that educators, parents and health

authorities will be able to make sleep a priority in the health education given to children, by making them understand how limiting their sleep can have consequences for their quality of life and health.

## Supporting information

**S1 File. Human participants research checklist.**
(PDF)

## Acknowledgements

The authors are indebted to all of the participating children and adolescents and their parents, and to the teachers who accommodated the survey during their classes.

The authors are also grateful to the French Ministry of Education, Direction générale de l'enseignement scolaire (DGESCO), direction des études de la performance et de la prospective (DEPP) and Office français de lutte contre les drogues et toxicomanie (OFDT).

The sleep survey is only part of this general survey, whose results are provided online in French at http://www.enclass.fr/index.php/publications-enclass

## Author contributions

**Conceptualization:** Damien LEGER, Stanislas SPILKA, Emmanuelle GODEAU.

**Data curation:** Emmanuelle GODEAU.

**Formal analysis:** Damien LEGER, Virginie EHLINGER, Stanislas SPILKA, Olivier LE-NEZET, Emmanuelle GODEAU.

**Funding acquisition:** Stanislas SPILKA, Emmanuelle GODEAU.

**Investigation:** Damien LEGER, Emmanuelle GODEAU.

**Methodology:** Damien LEGER, Virginie EHLINGER, Stanislas SPILKA, Olivier LE-NEZET, Emmanuelle GODEAU.

**Project administration:** Emmanuelle GODEAU.

**Resources:** Emmanuelle GODEAU.

**Supervision:** Damien LEGER, Emmanuelle GODEAU.

**Validation:** Virginie EHLINGER, Stanislas SPILKA, Brigitte FAUROUX, Victor PITRON, Emmanuelle GODEAU.

**Visualization:** Damien LEGER, Olivier LE-NEZET.

**Writing – original draft:** Damien LEGER, Emmanuelle GODEAU.

**Writing – review & editing:** Damien LEGER, Virginie EHLINGER, Stanislas SPILKA, Brigitte FAUROUX, Victor PITRON, Emmanuelle GODEAU.

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
