## [Decision Letter · Decision Letter 0]

21 Aug 2024

PONE-D-24-21580Two surveys separated by almost a decade reveals the inexorable decline in teenagers’ sleep time.PLOS ONE

Dear Dr. LEGER,

Thank you for submitting your manuscript to PLOS ONE. After careful consideration, we feel that it has merit but does not fully meet PLOS ONE’s publication criteria as it currently stands. Therefore, we invite you to submit a revised version of the manuscript that addresses the points raised during the review process.

We look forward to receiving your revised manuscript.

Kind regards,

Christian Veauthier, M.D.

Academic Editor

PLOS ONE

Journal Requirements:

"The authors are grateful to the French Ministry of Education (direction générale de l’enseignement 

scolaire, DGESCO ; direction des études de la performance et de la prospective, DEPP), Santé Publique 

France and OFDT. The authors are indebted to all of the participating children and adolescents and 

their parents, and to the teachers who accommodated the survey during their classes."

"The authors are grateful to the French Ministry of Education (direction générale de l’enseignement scolaire, DGESCO ; direction des études de la performance et de la prospective, DEPP), Santé Publique France and OFDT."

Reviewers' comments:

Reviewer's Responses to Questions

**Comments to the Author**

1. Is the manuscript technically sound, and do the data support the conclusions?

Reviewer #1: Yes

Reviewer #2: Yes

2. Has the statistical analysis been performed appropriately and rigorously? 

Reviewer #1: Yes

Reviewer #2: Yes

3. Have the authors made all data underlying the findings in their manuscript fully available?

Reviewer #1: Yes

Reviewer #2: Yes

4. Is the manuscript presented in an intelligible fashion and written in standard English?

Reviewer #1: Yes

Reviewer #2: Yes

5. Review Comments to the Author

Reviewer #1: The authors present results of an interesting epidemiological study on the temporal dynamics of sleep behavior in French schoolchildren. The methodology is good and the results are solid. My only concern is about the English in this manuscript.

For example, the following sentence sound strange for me:

"From October 1 2010 to June 30 2011, for the first time, we documented in this Journal the sleep of

French adolescents using the Health Behaviour in School-aged Children (HBSC/WHO) survey [5-6] and

the 2011 (April 1 to June 30) European School Survey Project on Alcohol and Other Drugs (ESPAD)

survey"

What do you mean by saying "...we documented in this Journal"?

I recommend having the manuscript revised linguistically by a native speaker.

Reviewer #2: The study compares data from a large representative survey conducted 8 years apart on juvenile sleep behavior and sleep parameters in France. The data is highly interesting and relevant to current study questions. I just have a few comments

Abstract

The term "total sleep time" is typically utilized in the context of polysomnography, where it is defined in a specific manner. This term is employed throughout the manuscript, indicating the presence of objective sleep data that were not collected in this study. I suggest the term sleep duration instead.

The terms sleep debt and too short sleep should already be defined in the methods section.

Results: The term “lost” is too interpretative. “Increase of 36.8. points” is not understandable because it was not explained beforehand.

Conclusion: A forward-looking conclusion would be desirable here.

Introduction

“These activities are a major reason.....” Please provide one or more literature sources for this statement.

“The statistical department of the French Ministry....the international requirements” Please provide a source here.

Statistical analyses

“...using National data...” Please indicate reference.

Results

“Impressive rates.....” The word is a little tendentiuos

Discussion

“To our knowledge...............” This sentence is somewhat confusing, as it is followed by a list of various studies on precisely this topic

Regarding the limitations

One limitation is certainly that the subjective quality of sleep was not assessed. It would be interesting to know whether the students also perceived their sleep as too short.

Additionally, the last sentence of the conclusion could be more specific to the results of this study, as it is currently formulated in a way that applies to all possible results in different areas.

6. PLOS authors have the option to publish the peer review history of their article (what does this mean? ). If published, this will include your full peer review and any attached files.

**Do you want your identity to be public for this peer review?** For information about this choice, including consent withdrawal, please see our Privacy Policy .

Reviewer #1: No

Reviewer #2: No

---

## [Author Response · Author response to Decision Letter 1]

11 Oct 2024

Dear Editor,

Thank you for all coments and revision points we carefully adressed as follows:

PONE-D-24-21580: Inexorable decline of sleep in teens: Answer to reviewers

We carefully checked, and followed the instructions.

Done

"The authors are grateful to the French Ministry of Education (direction générale de l’enseignement

scolaire, DGESCO ; direction des études de la performance et de la prospective, DEPP), Santé Publique

France and OFDT. The authors are indebted to all of the participating children and adolescents and

their parents, and to the teachers who accommodated the survey during their classes."

"The authors are grateful to the French Ministry of Education (direction générale de l’enseignement scolaire, DGESCO ; direction des études de la performance et de la prospective, DEPP), Santé Publique France and OFDT."

OK agree

All relevant data are within the manuscript.

There is no restriction.

Table 2 supplementary is positioned at the end of the manuscript.

The reference list is revised.

Reviewers' comments:

Reviewers' comments:

Reviewer's Responses to Questions

Comments to the Author

1. Is the manuscript technically sound, and do the data support the conclusions?

Reviewer #1: Yes

Reviewer #2: Yes

2. Has the statistical analysis been performed appropriately and rigorously?

Reviewer #1: Yes

Reviewer #2: Yes

3. Have the authors made all data underlying the findings in their manuscript fully available?

Reviewer #1: Yes

Reviewer #2: Yes

4. Is the manuscript presented in an intelligible fashion and written in standard English?

Reviewer #1: Yes

Reviewer #2: Yes

5. Review Comments to the Author

Reviewer #1: The authors present results of an interesting epidemiological study on the temporal dynamics of sleep behavior in French schoolchildren. The methodology is good and the results are solid. My only concern is about the English in this manuscript.

For example, the following sentence sound strange for me:

"From October 1 2010 to June 30 2011 for the first time, we documented in this Journal the sleep of French adolescents using the Health Behaviour in School-aged Children (HBSC/WHO) survey [5-6] and the 2011 (April 1 to June 30) European School Survey Project on Alcohol and Other Drugs (ESPAD) survey".

What do you mean by saying "we documented in this Journal”?

R: We wanted to say, “we published in Plos-One”. We revised it:

“Based on data collected from October 1, 2010 to June 30, 2011, we document here for the first time, we documented in this Journal the sleep of French adolescents using the Health Behaviour in School-aged Children (HBSC/WHO) survey [5-6] and the 2011 (April 1 to June 30) European School Survey Project on Alcohol and Other Drugs (ESPAD) survey [6]”

I recommend having the manuscript revised linguistically by a native speaker.

The manuscript has been revised by a native speaker (see all corrections on the redline manuscript, tables and highlights.

Reviewer #2: The study compares data from a large representative survey conducted 8 years apart on juvenile sleep behavior and sleep parameters in France. The data is highly interesting and relevant to current study questions. I just have a few comments

Abstract

The term "total sleep time" is typically utilized in the context of polysomnography, where it is defined in a specific manner. This term is employed throughout the manuscript, indicating the presence of objective sleep data that were not collected in this study. I suggest the term sleep duration instead.

In the method part, we concisely define total sleep time TST based on meticulous sleep logs as the difference between the time at which the participant went to bed and the time of day that they woke up, discounting the time needed to fall asleep. This definition is indeed close from the one obtained with objective polysomnography. We believe that sleep duration is too vague and may be confused with the subjective feeling of participants. This is why we propose to keep TST.

The terms sleep debt and too short sleep should already be defined in the methods section.

Thank-you, we introduced the definitions in the abstract method part: Sleep debt as was defined as a difference between the total sleep time (TST) during schooldays (TSTS) non-schooldays (weekends or vacations; TSTN) of over 2 hours. Too short sleep was assesses when TSTS was < 7 hours.

Results: The term “lost” is too interpretative. “Increase of 36.8. points” is not understandable because it was not explained beforehand.

Agree, we replace it by “the TSTS of middle schoolers decreases by an average of 20 minutes of sleep per night on weekdays, dropping from 8 h 35 min to 8 h 14 min.”

Conclusion: A forward-looking conclusion would be desirable here.

Thank-you, we replaced the conclusion by : “Faced with this trend, teachers and parents need to take preventive action to avoid an inexorable decline in teenagers' sleep.”

Introduction

“These activities are a major reason.....” Please provide one or more literature sources for this statement.

Thank-you, we added these two references:

Owens J; Adolescent Sleep Working Group; Committee on Adolescence. Insufficient sleep in adolescents and young adults: an update on causes and consequences. Pediatrics. 2014 Sep;134(3):e921-32.

Dresp-Langley B, Hutt A. Digital Addiction and Sleep. Int J Environ Res Public Health. 2022 Jun 5;19(11):6910. doi: 10.3390/ijerph19116910.

“The statistical department of the French Ministry...the international requirements” Please provide a source here.

Done.

Statistical analyses

“...using National data...” Please indicate reference.

Done.

Results

“Impressive rates.....” The word is a little tendentious

Agree replaced by “high”.

Discussion

“To our knowledge...............” This sentence is somewhat confusing, as it is followed by a list of various studies on precisely this topic

Agree, the sentence was replaced by “Our study was specifically designed to assess the epidemiology of TST in a large number of children and adolescents”

Regarding the limitations

One limitation is certainly that the subjective quality of sleep was not assessed. It would be interesting to know whether the students also perceived their sleep as too short.

Agree we have added this limitation: “Third, we did not interview children on how they perceived their sleep was too short or not.”

Additionally, the last sentence of the conclusion could be more specific to the results of this study, as it is currently formulated in a way that applies to all possible results in different areas.

Thank you, we replace the last sentence by this most focused one: “Given these results, we hope that educators, parents and health authorities will be able to make sleep a priority in the health education given to children, by making them understand how limiting their sleep can have consequences for their quality of life and health.”

---

## [Decision Letter · Decision Letter 1]

18 Nov 2024

Two surveys separated by almost a decade reveal the inexorable decline in sleep time in teens.

PONE-D-24-21580R1

Dear Dr. LEGER,

We’re pleased to inform you that your manuscript has been judged scientifically suitable for publication and will be formally accepted for publication once it meets all outstanding technical requirements.

Kind regards,

Christian Veauthier, M.D.

Academic Editor

PLOS ONE

Additional Editor Comments (optional):

Reviewers' comments:

Reviewer's Responses to Questions

**Comments to the Author**

1. If the authors have adequately addressed your comments raised in a previous round of review and you feel that this manuscript is now acceptable for publication, you may indicate that here to bypass the “Comments to the Author” section, enter your conflict of interest statement in the “Confidential to Editor” section, and submit your "Accept" recommendation.

Reviewer #1: All comments have been addressed

2. Is the manuscript technically sound, and do the data support the conclusions?

Reviewer #1: Yes

3. Has the statistical analysis been performed appropriately and rigorously? 

Reviewer #1: Yes

4. Have the authors made all data underlying the findings in their manuscript fully available?

Reviewer #1: Yes

5. Is the manuscript presented in an intelligible fashion and written in standard English?

Reviewer #1: Yes

6. Review Comments to the Author

Reviewer #1: The improvements adressed all questions, which I had, in an appropriate manner. I recommend this manuscript for publication.

7. PLOS authors have the option to publish the peer review history of their article (what does this mean? ). If published, this will include your full peer review and any attached files.

**Do you want your identity to be public for this peer review?** For information about this choice, including consent withdrawal, please see our Privacy Policy .

Reviewer #1: No

---

## [Editor Report · Acceptance letter]

PONE-D-24-21580R1

PLOS ONE

Dear Dr. LEGER,

I'm pleased to inform you that your manuscript has been deemed suitable for publication in PLOS ONE. Congratulations! Your manuscript is now being handed over to our production team.

Kind regards,

on behalf of

Dr. Christian Veauthier

Academic Editor

PLOS ONE